# Through-bond and through-space radiofrequency amplification by stimulated emission of radiation
Ivan A. Trofimov [1,2] ✉, Oleg G. Salnikov [2] ✉, Andrey N. Pravdivtsev [3], Henri de Maissin [1,4], Anna P. Yi[2,5], Eduard Y. Chekmenev [6], Jan-Bernd Hövener [3], Andreas B. Schmidt [1,4,6] & Igor V. Koptyug [2]

Radio Amplification by Stimulated Emission of Radiation (RASER) is a phenomenon observed during nuclear magnetic resonance (NMR) experiments with strongly negatively polarized systems. This phenomenon may be utilized for the production of very narrow NMR lines, background-free NMR spectroscopy, and excitation-free sensing of chemical transformations. Recently, novel methods of producing RASER by ParaHydrogen-Induced Polarization (PHIP) were introduced. Here, we show that pairwise addition of parahydrogen to various propargylic compounds induces RASER activity of other protons beyond those chemically introduced in the reaction. In high-field PHIP, negative polarization initiating RASER is transferred via intramolecular cross-relaxation. When parahydrogen is added in Earth's field followed by adiabatic transfer to a high field, RASER activity of other protons is induced via both J-couplings and cross-relaxation. This through-bond and through-space induction of RASER holds potential for the ongoing development and expansion of RASER applications and can potentially enhance spectral resolution in two-dimensional NMR spectroscopy techniques.

Under certain conditions, stimulated emission of radiation may lead to the amplification of specific frequencies[1]. This phenomenon is universally applied with the most common example being the amplification of waves with frequencies corresponding to visible, IR and UV light (LASER)[2]. The less common variants of such phenomena are amplifications of micro- and radio waves: MASER[1] and radio amplification by stimulated emission of radiation (RASER)[3–7], respectively. Recently, the RASER effect became a point of interest attraction in the nuclear magnetic resonance (NMR) community because it allowed to produce dramatically narrowed spectral lines with full width at half maximum (FWHM) of several mHz[7,8] and to detect NMR signals without an application of a radiofrequency (RF) pulse which can be used for background-free NMR spectroscopy[9], sensing of chemical transformations[10,11] and even magnetic resonance imaging (MRI)[12].

RASER requires strong negative polarization of nuclear spin systems and interaction with the resonator with high quality factor to produce the stimulated emission[13]. RASER efficiency is characterized by the radiation damping rate $1/\tau_{RD}$:

$$1/\tau_{RD} = -(\mu_0/2)\eta Q\gamma M_0 = -(\mu_0/4)\eta Q\gamma^2\hbar n_S P \tag{1}$$

where $\mu_0$ is vacuum permeability, $\eta$ is the coil filling factor, $Q$ is the resonator quality factor, $\gamma$ is the gyromagnetic ratio, $M_0$ is the initial magnetization, $\hbar$ is the reduced Planck constant, $n_S$ is the spin density, and $P$ is the initial nuclear spin polarization. The nuclear spin ensemble must be negatively polarized ($P < 0$) for RASER to occur since only in this case the radiation damping rate is positive.

The RASER arises when the following condition is fulfilled:

$$1/\tau_{RD} > 1/T_2^* + 1/\tau_p \tag{2}$$

where $1/T_2^* = 1/T_2 + \gamma\Delta B_0$ is the transverse relaxation rate (in the case of linear field gradients), and $1/\tau_p$ is the polarization pumping rate. Thus, one can increase concentrations or polarization of the compounds, improve the

[1]Division of Medical Physics, Department of Diagnostic and Interventional Radiology, University Medical Center Freiburg, Faculty of Medicine, University of Freiburg, Freiburg, 79106, Germany. [2]International Tomography Center SB RAS, 3A Institutskaya St., 630090 Novosibirsk, Russia. [3]Section Biomedical Imaging, Molecular Imaging North Competence Center (MOIN CC), Department of Radiology and Neuroradiology, University Medical Center Schleswig-Holstein and Kiel University, 24118 Kiel, Germany. [4]German Cancer Consortium (DKTK), partner site Freiburg and German Cancer Research Center (DKFZ), 280 Im Neuenheimer Feld, Heidelberg, 69120, Germany. [5]Novosibirsk State University, 2 Pirogova St., 630090 Novosibirsk, Russia. [6]Department of Chemistry, Integrative Biosciences (Ibio), Karmanos Cancer Institute (KCI), Wayne State University, Detroit, Michigan, 48202, USA. ✉e-mail: ivan.trofimov@uniklinik-freiburg.de; salnikov@tomo.nsc.ru

quality of the resonator[7,14], or decrease the transverse relaxation rate or polarization pumping rate to ease the requirements for the RASER emission. Given that there is a source of continuous polarization pumping, RASER may achieve a steady state that continuously produces strong NMR signals[7,15].

The strong negative nuclear spin polarization required to meet RASER conditions (Eq. 2) can be achieved using hyperpolarization techniques[16–18]. RASER was demonstrated using several of these techniques: spin-exchange optical pumping of noble gases[3,4,6], dynamic nuclear polarization (DNP)[8,19–21], and parahydrogen-based techniques such as parahydrogen-induced polarization (PHIP)[10,11,22–25] and signal amplification by reversible exchange (SABRE)[7,13,24,26]. RASER emissions may be also obtained by inversion of thermal polarization provided it is sufficiently high[27,28].

A wide range of substances hyperpolarized in PHIP experiments were observed to yield RASER (commonly referred to as PHIP RASER)[10,11,22–25]. The PHIP technique is based on the addition of parahydrogen ($p$-$H_2$, singlet state of molecular hydrogen) to an unsaturated precursor[29–31]. To produce such a product with high nuclear spin polarization, it is important that the added protons $H_A$ and $H_B$ originate from the same $p$-$H_2$ molecule (this is referred to as "pairwise addition") and are magnetically inequivalent[29]. Depending on the strength of magnetic field $B_0$ in which hydrogen addition is conducted, one may observe two types of NMR signals: parahydrogen and synthesis allow dramatically enhanced nuclear alignment (PASADENA) at high-field conditions ($|J_{AB}| \ll \gamma\Delta\delta B_0$)[29,30] or adiabatic longitudinal transport after dissociation engenders net alignment (ALTADENA) at low-field conditions ($|J_{AB}| \gg \gamma\Delta\delta B_0$)[32]. Here $J_{AB}$ is the spin-spin coupling constant between $H_A$ and $H_B$, $\Delta\delta$ is the chemical shift difference between the NMR signals of $H_A$ and $H_B$. Both PASADENA and ALTADENA techniques are compatible with RASER[15,22,23].

Interestingly, PHIP RASER may induce a spontaneous transfer of polarization via intermolecular dipolar interactions to other analytes, which are not chemically interacting with the hyperpolarized molecule. This effect is called parahydrogen and RASER-induced nuclear Overhauser effect (PRINOE)[33,34].

Previously [1]H PHIP RASER was observed for molecules that did not contain any additional groups that have significant $J$-coupling with the $H_A$ and $H_B$ protons, such as ethyl acetate or 2-hydroxyethyl propionate. As a result, bimodal RASER with the two RASER-active frequencies corresponding to $H_A$ and $H_B$ protons was observed[24]. Continuous PHIP RASER was observed for methyl acrylate that contains additional vinyl proton coupled to $H_A$ and $H_B$, however, it was not active in the produced RASER[22]. Due to the increasing number of possible implementations of PHIP RASER, it is important to study the characteristics of PHIP RASER in more complex multispin systems, which can potentially produce multiple RASER-active modes.

In this work, we studied PHIP RASER characteristics of hyperpolarized (HP) allylic and substituted allylic proton systems and phenomena that arise during these experiments and found that, along with inducing RASER via scalar coupling network, PHIP RASER also induces intramolecular PRI-NOE with subsequent RASER emission of the polarization receiving nuclei.

## Results and discussion
### Experimental considerations
Two different approaches for the creation of hyperpolarization sufficient for the PHIP RASER effect to arise may be used: generation of stationary negative polarization of the sample via continuous slow bubbling of $p$-$H_2$[22] or creation of initial negative hyperpolarization that is then allowed to relax[23]. In this work, the second approach was implemented.

Propargylic compounds 1'–4' were selected as the PHIP precursors because corresponding hydrogenation product compounds 1–4 (Fig. 1a) have similar $J$-coupling networks (see Supplementary Note 1). Of note, hyperpolarization of compounds 1 and 2 via PHIP was previously investigated at conditions far from RASER[35,36]. To produce HP 1–4, we conducted pairwise addition of $p$-$H_2$ to 1'–4' over [Rh(nbd)(dppb)]BF$_4$ catalyst ([Rh], nbd = norbornadiene, dppb = 1,4-bis(diphenylphosphino)butane). In order to achieve initial magnetization large enough to initiate the RASER activity precursors 1'–4' were used in a high concentration of 800 mM. The [Rh] catalyst was produced in situ by the reaction of [Rh(nbd)$_2$]BF$_4$ and dppb in the methanol-d$_4$ solution.

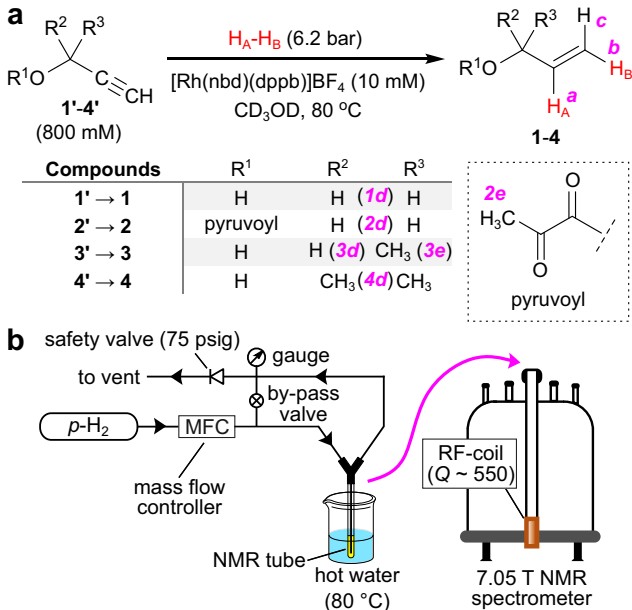

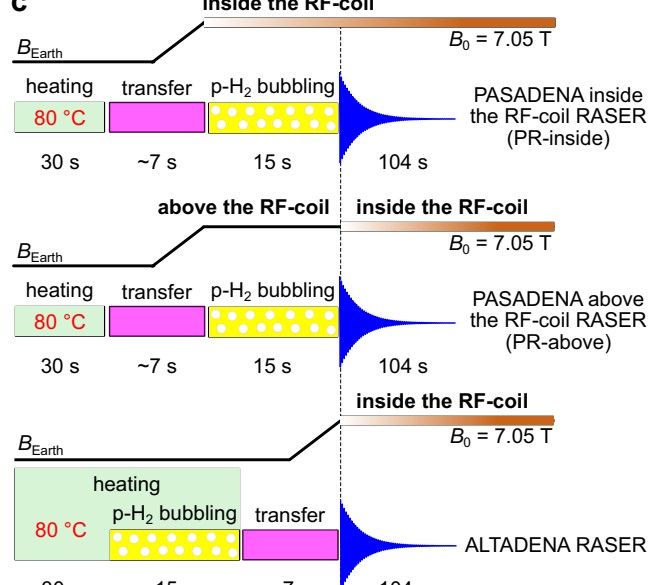

**Fig. 1 | Schematic representation of the reactions, setup and experimental protocols used to study RASER activity.** **a** Reaction scheme of pairwise addition of $p$-$H_2$ to precursors 1'–4' to yield HP products 1–4 (R[1]–R[3] are different substituent groups for propargylic/allylic fragment, see table below the reaction scheme): allyl alcohol (1), allyl pyruvate (2), 3-buten-2-ol (3), 2-methyl-3-buten-2-ol (4) (corresponding precursors are propargyl alcohol (1'), propargyl pyruvate (2'), 3-butyn-2-ol (3'), 2-methyl-3- butyn-2-ol (4')). Red $H_A$ and $H_B$ atoms stand for parahydrogen-nascent atoms. Magenta letters denote different types of protons in the molecules 1–4. **b** Scheme of the experimental setup used for RASER experiments. **c** Schematic event sequences in PR-inside (PASADENA inside the RF-coil RASER), PR-above (PASADENA above the RF-coil RASER), and ALTADENA RASER experiments. [1]H NMR signals were acquired for 104 s without the application of any RF excitation pulses.

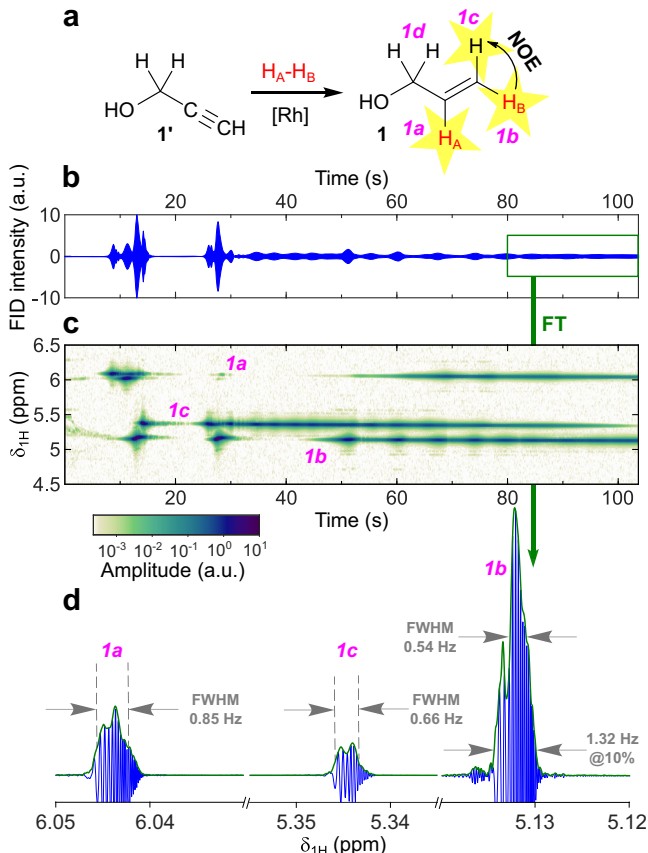

**Fig. 2 | PASADENA RASER of 1 demonstrates activity of the 1c proton induced by intramolecular cross-relaxation. a** Reaction scheme of hydrogenation of **1'** with $p$-$H_2$ to produce HP **1**. Red $H_A$ and $H_B$ atoms stand for parahydrogen-nascent atoms; protons labelled by yellow stars are RASER-active. RASER activity of *1c* protons is induced via Nuclear Overhauser Effect (NOE). **b** $^1H$ RASER signal acquired during PR-inside protocol experiment with **1'**. Arbitrary units (a.u.) denoting FID (free induction decay) signal intensity are consistent in all graphs presented in this work (including Supplementary Information). **c** Observed evolution of $^1H$ NMR signals produced from RASER signal presented in panel (**b**). The amplitude color scheme and arbitrary units (a.u.) are consistent in all spectrographs of $^1H$ NMR signals presented in this work. For the algorithm to produce the spectrographs see Supplementary Methods. **d** $^1H$ NMR spectrum produced by FT (Fourier transform) of 80–104 s timeframe of the $^1H$ RASER signal (green and blue graphs represent magnitude and real values, respectively); all three protons produce narrow NMR signals (@10%—full width at 10% of magnitude).

Hydrogenation of precursors **1'–4'** was performed via bubbling of $p$-$H_2$ (~94% enrichment) at 6.2 bar for 15 s (Fig. 1b). Before hydrogenation, the sample was pressurized and then preheated at 80 °C for 30 s at the Earth's magnetic field (~50 µT). Depending on the particular experimental protocol (Fig. 1c), hydrogenation was conducted either directly after preheating of the sample while the temperature was maintained (ALTADENA RASER) or after the preheating and subsequent transfer to the 7.05 T field of the NMR spectrometer (PASADENA RASER protocols, see details below). Acquisition of $^1H$ NMR spectra was performed by opening the detector of the NMR probe ($Q ≈ 550$) without the application of RF pulses. More details can be found in the Methods and Supplementary Methods.

## PASADENA RASER (PR)

As the RASER effect is a result of an interaction of an RF coil with a hyperpolarized spin ensemble, we compared two variants of hyperpolarization preparation at PASADENA conditions (Fig. 1c). In the first variant, the preheated sample is placed inside the RF coil followed by hydrogenation of the precursor after which the NMR signal acquisition is initiated (PR-inside protocol). In the second variant (PR-above protocol), the preheated

sample with an unsaturated precursor is positioned above the RF coil but at the same magnetic field of 7.05 T followed by $p$-$H_2$ bubbling. The difference between the two PASADENA RASER protocols is that in the PR-*above* protocol strong PASADENA polarization is first accumulated and then it starts interacting with the coil while in the PR-*inside* protocol these two processes occur simultaneously.

Implementation of the PR-inside protocol for all precursors **1'–4'** resulted in RASER of HP products **1–4**. In the case of **1**, all three frequencies *1a–1c* corresponding to the vinyl protons were prominent (Fig. 2b, c): RASER was long-lived, being active until the end of signal acquisition. Note that in the interval from ~50 to 100 s all three protons give rise to the RASER signal. Repeating this experiment on a different day produced RASER with only *1c* and *1a* being RASER-active (Supplementary Fig. 2a, b), indicating that even slight variations in experimental conditions can significantly change RASER appearance.

HP products **2** and **3** obtained in PR-inside experiments produced RASER at the frequencies of all three vinyl protons *a–c* (Supplementary Fig. 2c–f). In the case of HP **3**, the signal *3a* was represented by two components; such behavior was previously observed for RASER signals with the distance between the components of a signal being close or equal to the $J$-coupling constant[23,34]. In our experiment, however, the distance between the components was ~23 Hz instead of $J_{3ac} ~ 17.15$ Hz expected for the vinyl system in molecule **3** (Supplementary Fig. 2g). The reason is that this distance is represented not by a single $J$-coupling but by a combination of two, as the sum $J_{3ac} + J_{3ad} ~ 23$ Hz in this case corresponds to the distance between two negatively polarized spectral lines in the PASADENA signal of *3a* protons.

HP **4** produced from **4'** at PASADENA conditions yielded long-lived RASER with three prominent frequencies *4a–4c* appearing at different moments of time (Fig. 3). Two patterns of equidistant satellite signals (usually referred to as *frequency combs*) are prominent in the resulting spectrograph during separate timeframes of ca. 20–50 s and ca. 60–90 s. It is important to note that the distances $\zeta_{ij}$ between these signals are not exactly equal to the differences in chemical shifts $\Delta\delta_{ij}$ between the corresponding peaks in thermal $^1H$ NMR spectra (Supplementary Table 2, $\Delta\delta_{ac} = 0.79$ ppm (238 Hz), $\Delta\delta_{bc} = 0.23$ ppm (69 Hz), $\Delta\delta_{ab} = 1.02$ ppm (307 Hz)) but are very close to them. Analysis of the corresponding PASADENA spectrum of the relaxing sample (see Supplementary Fig. 6d) showed that the distances $\zeta_{ij}$ perfectly match the distances between the emissive components of the corresponding antiphase multiplets *4a–4c*.

Usually, when the addition of $p$-$H_2$ is conducted at PASADENA conditions, the only HP $^1H$ NMR signals to be observed correspond to the protons that came from $p$-$H_2$ (denoted further as $H_A$ and $H_B$). This is because at high field the interaction of nuclear spins with magnetic field $B_0$ is much stronger than the scalar interaction between different nuclear spins. However, in our experiments, we also detected the RASER signals of the $c$ protons. The reasons for this may be cross-relaxation (NOE)[37], [Rh]-mediated proton exchange during hydrogenation[38] or side processes of *trans*-hydrogenation (when $p$-$H_2$-derived H atoms are added in positions $a$ and $c$ rather than $a$ and $b$).

To investigate the feasibility of proton exchange and *trans*-hydrogenation, the compounds **1'** and **4'** were introduced in reaction with $D_2$ (see Supplementary Note 9). The acquired $^1H$ and $^2H$ NMR spectra of the reaction products (Supplementary Fig. 13) showed that *trans*-addition does not happen to any noticeable extent, while hydrogen exchange leading to the product with three deuterons (occupying all positions $a$, $b$ and $c$) is possible in principle. However, it is a minor side process, as it corresponds only to ca. 3% of formed product molecules. We thus attribute the PASADENA RASER of $c$ protons to the action of intramolecular PRINOE effects. This could occur via NOE or be induced by RASER. In the latter case, RASER applied on two protons can fulfill Hartmann–Hahn condition[39]–RASER emissions produced by HP molecules in the solution may trigger polarization transfer from $b$ to $c$ protons akin to RF-irradiation triggering homo- and heteronuclear polarization transfer[40]. The observed emissive lines for the $c$ signals in the $^1H$ NMR spectra of the relaxing samples support the former explanation (see Supplementary Notes 7 and 8 for further details).

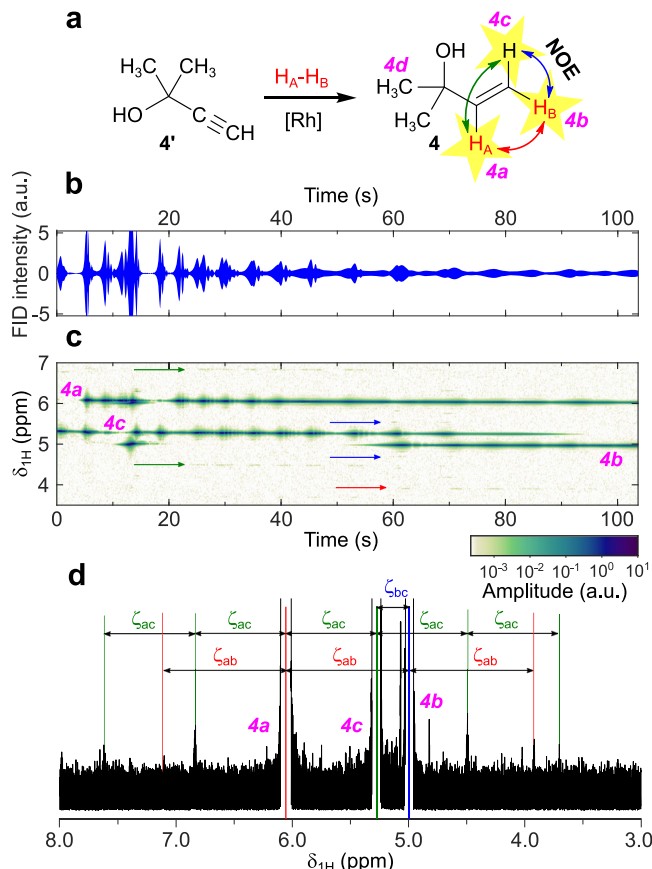

**Fig. 3 | PASADENA RASER of 4 demonstrates complex interplay of RASER-active frequencies. a** Reaction scheme of hydrogenation of **4'** with *p*-H$_2$ to produce HP **4**. Red H$_A$ and H$_B$ atoms stand for parahydrogen-nascent atoms; protons labelled by yellow stars are RASER-active. RASER activity of *4c* protons is induced via Nuclear Overhauser Effect (NOE). Colored arrows connect the pairs of protons between which the corresponding distances $\zeta_{ij}$ in the spectrum (**d**) are presented. **b** $^1$H RASER signal acquired during PR-inside protocol experiment with **4'**. The RASER signals are zoomed in vertical dimension so that 2% of all acquired RASER data points with extreme FID intensities are clipped; thus, the initial strong bursts are not visible fully but the dynamic of weaker signals is articulated. **c** Observed evolution of $^1$H NMR signals produced from RASER signal presented in panel (**b**). Frequency combs are marked by the arrows of corresponding colors. **d** $^1$H NMR spectrum produced by FT of RASER signal presented in panel (**b**); the spectrum is displayed as magnitude values and is zoomed to the baseline. Values $\zeta_{ij}$ are $\zeta_{ac}$ = 0.78 ppm (235 Hz), $\zeta_{bc}$ = 0.29 ppm (86 Hz), $\zeta_{ab}$ = 1.07 ppm (320 Hz).

To establish the exact nature of the occurring polarization transfer experiments involving a *Q*-switchable NMR probe may be conducted.

It is important to note that acquired PASADENA RASER signals last over 100 s, which is at least 3 times longer than the earlier reported PASADENA RASER of ethyl acetate and 2-hydroxyethyl propionate (ca. 25–30 s)[23,24]. We attribute this difference in longevity of RASER signals mainly to the fact that after the end of bubbling of *p*-H$_2$ there is still a considerable amount of both precursor and dissolved *p*-H$_2$ left in the solution. Hence, during the acquisition of RASER signal the reaction of the precursor with residual *p*-H$_2$ is still ongoing, albeit slower than during the active supply of *p*-H$_2$. The chemical conversion values were deduced for each RASER experiment from $^1$H NMR spectra of the solutions acquired after the relaxation of spin polarization (see Supplementary Note 2). In contrast, in the works mentioned above HP products were obtained quantitatively from the precursors. Thus, the ongoing process of hydrogenation is quite important for the longevity of the obtained RASER signals.

Comparative PR-above experiments revealed two significantly different patterns of RASER bursts (Supplementary Note 5). RASER signals produced by HP products **2**–**4** consisted of separate gradually diminishing bursts of *2b*–*4b* signals with additional bursts of *4c* for **4** (Supplementary Fig. 3c–h); this behavior is in full accordance with expectations for PASADENA RASER. On the other hand, the PASADENA RASER of **1** produced by PR-*above* protocol consisted of a very strong first burst of *1a* and *1b* lasting for 1 s (the burst also includes weak *1d* signal, most likely, due to cross-relaxation) and a separate RASER of *1c* signal emerging only after ca. 30 s and lasting over 1 min after that. This pattern of signals was successfully reproduced for HP product **1** during the PR-above experiment on another day, so this behavior is likely caused by higher chemical conversion of **1'** compared to **2'** and **3'**. Higher chemical conversion leads to larger molar polarization of nascent H$_A$ and H$_B$ protons which results in stronger PRINOE-mediated polarization of the *1c* proton. The emergence of the *4c* signal in the RASER of **4** supports this hypothesis, as chemical conversion in this experiment was also significant.

The results obtained in PR-above experiments are described well by the assumption that the main source of polarization after the stopping of the *p*-H$_2$ flow is the continuing pairwise addition of dissolved *p*-H$_2$ to the remaining unsaturated precursor, as we observe a series of diminishing bursts of RASER for HP products **2**–**4**. However, there is a more complex and prolonged signal dynamics in the case of PR-inside experiments. This may be explained by the fact that during PR-inside experiments, the nascent polarization of the sample interacts with the RF coil during polarization buildup and before the start of acquisition, while in PR-above experiments, the accumulated polarization starts interacting with the coil only after the hyperpolarization process is completed. Indeed, during the bubbling of *p*-H$_2$ the pumping rate is very high, leading to chaotic evolution of magnetization; bubbling also leads to constant displacement of a significant part of the solution out of the sensitive zone of the NMR probe, leading to significantly worsened field homogeneity. This was studied on the example of **1** continuously produced during the acquisition of the RASER signal (Supplementary Note 8); a series of irregularly positioned bursts of *1b* signals was observed (Supplementary Fig. 10). A series of RASER bursts of *4b* in fast succession was observed for **4** in similar conditions (Supplementary Fig. 12). The absence of the H$_A$ signals in this case may be rationalized by the fact that H$_B$ protons have fewer *J*-couplings than H$_A$ ones. Because of that, H$_B$ signal components reach the magnetization threshold faster than H$_A$ signal components and initiate RASER (spin order transformations happen, $|I_z S_z\rangle \rightarrow |I_z S_x\rangle$, where *I* and *S* represent nuclear spins of H$_A$ and H$_B$, respectively). The line shapes of *1a* and *1b* signals in the NMR spectra acquired during the relaxation of the corresponding sample resemble the $|I_z - S_z\rangle$ spin order (Supplementary Fig. 11). This demonstrates that H$_B$-initiated RASER further converts $|I_z S_z\rangle$ and $|I_z S_x\rangle$ spin orders into $|I_z - S_z\rangle$. Because H$_A$ is positively polarized its appearance in the RASER signal becomes impossible. Hence, the main reason for the chaoticity of PASADENA RASER produced in the PR-inside experiments is the initial high rate of *p*-H$_2$ bubbling while the sample is inside the NMR probe.

In several PASADENA RASER experiments, observed signals experienced a slight drift of chemical shift with evolution time. This drift may be explained by distant dipolar fields (DDF) effects: a polarized sample produces its own magnetic field, which relaxes over time, leading to the offset of NMR signals relative to the field of the spectrometer[6,41–43].

## ALTADENA RASER

The experiments employing the ALTADENA RASER protocol were conducted twice for each of the substrates **1'**–**4'**. The resulting $^1$H NMR spectrographs of RASER of HP products **1**–**4** acquired for 104 s are presented in Fig. 4. The main difference from PASADENA experiments is that among the two *p*-H$_2$-derived protons only H$_B$ proton leads to RASER as it is negatively polarized after magnetic field cycling (H$_A$ is positively polarized and hence does not induce RASER). Magnetic field cycling also resulted in polarization transfer to other protons besides vinyl moiety via the scalar couplings network (methylene protons *1d*–*2d*, methanetriyl proton *3d* and methyl protons *3e*). The arising polarization happens to be negative, leading to RASER activity. These protons were RASER-passive in the PASADENA experiments.

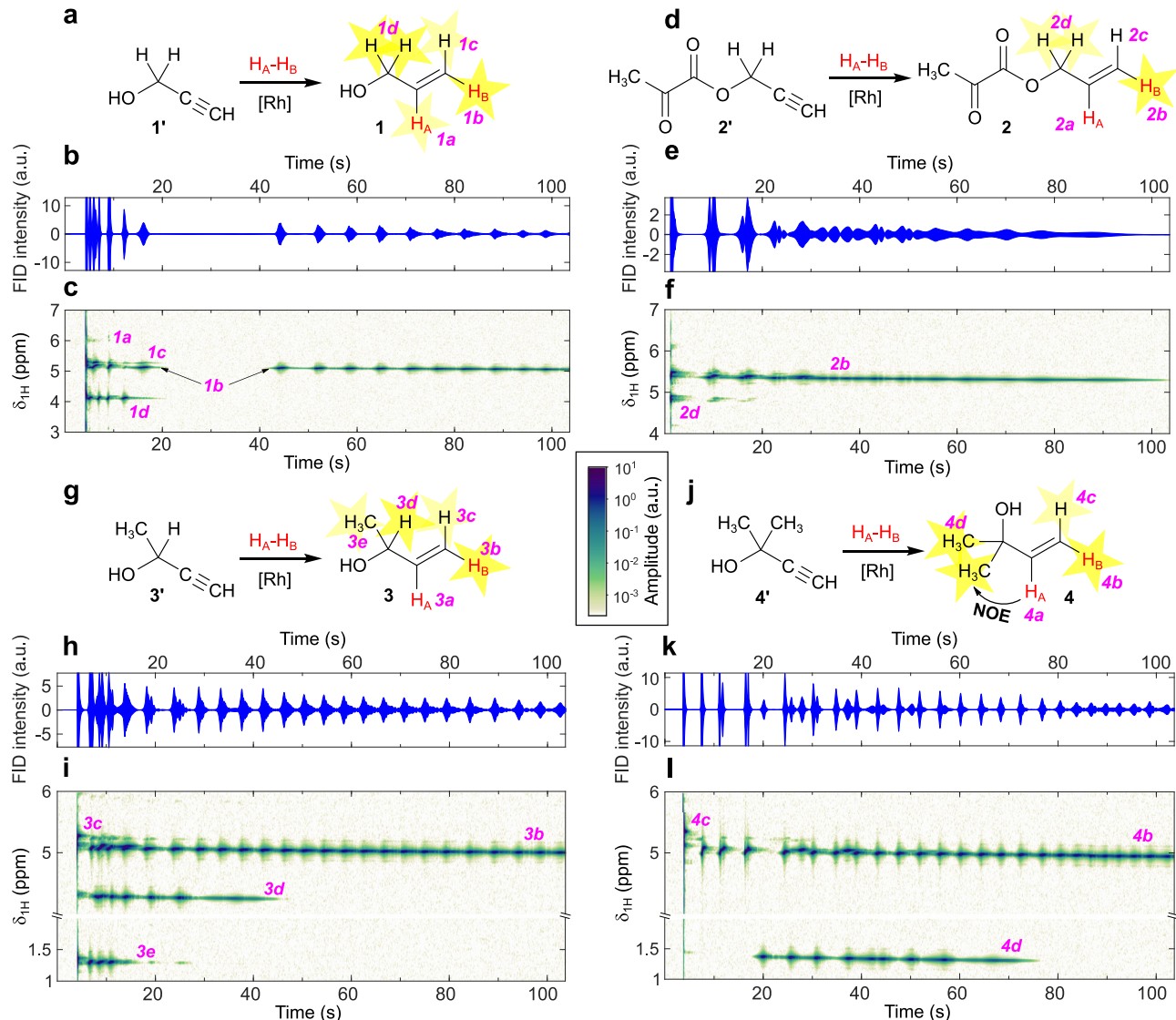

**Fig. 4 | ALTADENA RASER of propargylic compounds demonstrates RASER activity induced via *J*-couplings. a** Reaction scheme of hydrogenation of **1'** with *p*-H$_2$ to produce HP **1**. **b** $^1$H RASER signal acquired during ALTADENA RASER experiment with **1'**. **c** Observed evolution of $^1$H NMR signals produced from the RASER signal presented in panel (**b**). **d** Reaction scheme of hydrogenation of **2'** with *p*-H$_2$ to produce HP **2**. **e** $^1$H RASER signal acquired during ALTADENA RASER experiment with **2'**. **f** Observed evolution of $^1$H NMR signals produced from the RASER signal presented in panel (**e**). **g** Reaction scheme of hydrogenation of **3'** with *p*-H$_2$ to produce HP **3**. **h** $^1$H RASER signal acquired during ALTADENA RASER experiment with **3'**. **i** Observed evolution of $^1$H NMR signals produced from the RASER signal presented in panel (**h**). **j** Reaction scheme of hydrogenation of **4'** with *p*-H$_2$ to produce HP **4**. **k** $^1$H RASER signal acquired during ALTADENA RASER experiment with **4'**. **l** Observed evolution of $^1$H NMR signals produced from the RASER signal presented in panel (**k**). Red H$_A$ and H$_B$ atoms stand for parahydrogen-nascent atoms; protons labelled by yellow stars are RASER-active. The RASER signals are zoomed in vertical dimension so that 2% of all acquired RASER data points with extreme FID intensities are clipped in (**b, e, h, k**) FID panels. The amplitude colour bar for the (**c, f, i, l**) panels is provided in the middle of the Figure.

As all experiments were repeated at least twice, we can compare the reproducibility of the ALTADENA RASER effect. So, in one case HP product **1** produced a RASER signal consisting of several diminishing bursts of all HP protons with *1b* frequency reemerging and persisting until the end of the signal acquisition after 20 s of "radio silence" (Fig. 4b, c). During another experiment the same frequencies were active, but an intermittent pattern was observed for *1b* and *1d* frequencies in the first 40 s of RASER (Supplementary Fig. 4). After 45 s the frequencies entered the coexistence mode producing a complex continuous signal after 80 s. This inconsistency in patterns of produced signals may be explained by the chaotic behavior of the RASER system. Later in this Section we will also suggest an explanation for the long persistence of *1d* signal (Supplementary Fig. 4). Repeating the experiment with a prolonged acquisition time of 250 s allowed us to demonstrate that RASER signals of **1** are active for ~150 s after the beginning of acquisition (Supplementary Fig. 5).

The RASER produced by HP **2** obtained at ALTADENA conditions contained only two frequencies corresponding to terminal vinyl (*2b*) and methylene (*2d*) protons (Fig. 4e, f). ALTADENA RASER resulting from HP **3** contained four active frequencies corresponding to terminal vinyl (*3b* and *3c*), methanetriyl (*3d*), and methyl (*3e*) protons. The frequency corresponding to *3b* was active during the whole acquisition time, while *3d* and *3e* frequencies were active for a shorter duration of time immediately after insertion of the sample into the NMR probe.

Finally, in the case of **4**, the main RASER-active frequency corresponding to *4b* proton was observed during the whole acquisition time; however, at ~20 s after the acquisition was started, a new frequency corresponding to methyl protons (*4d*) emerged in the spectrum and persisted for ~50 s (Fig. 4k, l). Similar behavior was successfully reproduced during the experiment on another day. It appears that polarization transfer via intra-molecular NOE interactions between *4a* and *4d* protons takes place (see

Supplementary Note 10 for details). This PRINOE effect builds up negative polarization on *4d* protons, which becomes strong enough to exceed the RASER threshold. On this notice, a pronounced intermolecular PRINOE effect was also observed for the terminal alkyne protons of the precursor molecules at both PASADENA and ALTADENA conditions, which was discovered during the analysis of the ¹H NMR spectra of the relaxing samples acquired 10–20 s after the end of acquisition of RASER signal (Supplementary Note 7).

In all the spectrographs of ¹H NMR signals obtained in ALTADENA RASER experiments, significant drift of the chemical shifts with time was observed. In addition to the DDF effects, here there is an additional temperature shift because a hot solution (80 °C) right after the PHIP hyperpolarization process is inserted into the NMR spectrometer. This results in a gradual cooling of the sample during RASER signal acquisition. In the case of PASADENA RASER experiments, the samples were cooled down during *p*-H₂ bubbling so the temperature variations in the course of RASER signal acquisition can be neglected.

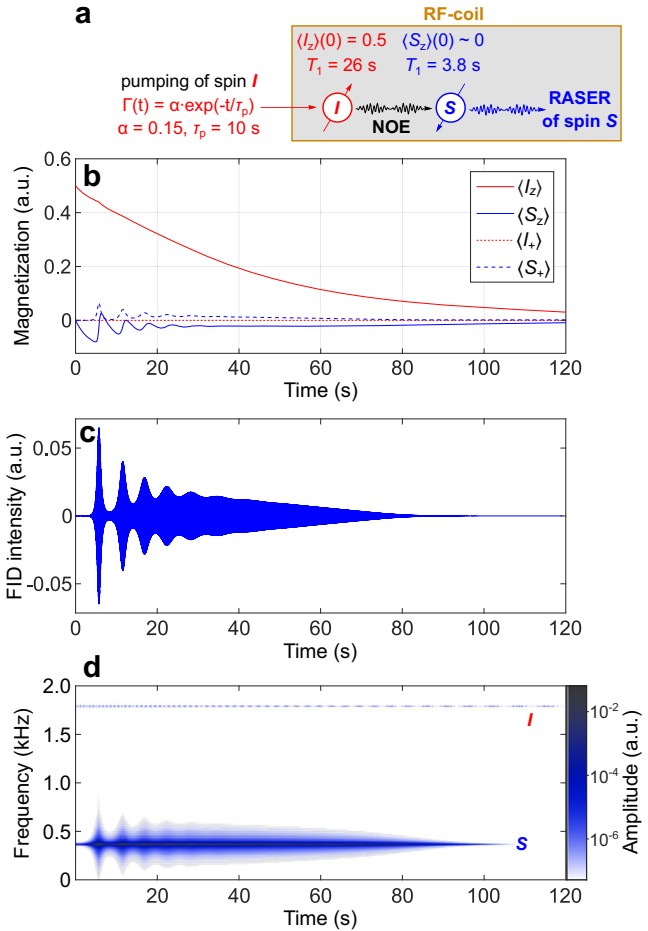

**Fig. 5 | Simulation of RASER induced via NOE interactions. a** Scheme of NOE transfer and main simulation parameters. **b** Evolution of longitudinal $\langle \hat{I}_z \rangle$ (solid red line), $\langle \hat{S}_z \rangle$ (solid blue line) and transverse $\langle \hat{I}_+ \rangle$ (dashed red line), $\langle \hat{S}_+ \rangle$ (dashed blue line) magnetization components of spins $I$ and $S$. **c** RASER signal obtained from the simulated $\langle \hat{I}_+ \rangle$ and $\langle \hat{S}_+ \rangle$ values. **d** Evolution of the NMR signals produced from the RASER signal presented in panel (**c**). Contribution of the spin $I$ in FID is evident but it is 4 orders of magnitude weaker than that of $S$ and would be below the noise level in the real experiment. The NMR parameters of *4a* and *4d* protons were assumed for $I$ and $S$, respectively. Parameters for the simulation are: $\rho_I = 1/26$ s$^{-1}$, $\rho_S = 1/3.81$ s$^{-1}$, $\sigma = 0.0641$ s$^{-1}$, $\tau_{RD} = 44.3$ ms (see Supplementary Note 3), $M_0 = \langle \hat{I}_z \rangle(0) = 0.5$, $\langle \hat{I}_+ \rangle(0) = I_z^{eq}$, $\langle \hat{S}_z \rangle(0) = \langle \hat{S}_+ \rangle(0) = S_z^{eq}$, $\nu_I = \nu_S = 1$ s$^{-1}$, $I_z^{eq} = S_z^{eq} = 2.4210^{-5}$ (numerically equal to thermal polarization of ¹H nuclei at 7.05 T magnetic field in absolute units).

Spectrographs of ¹H NMR signals produced in ALTADENA RASER experiments demonstrate initial bursts containing frequencies corresponding to all negatively polarized protons (for polarization signs see Supplementary Figs. 8 and 15). After that, signals corresponding to H$_B$ produce consecutive bursts along with other *J*-coupled protons (assigned as *1–3d* or *3e*) which eventually relax below the magnetization threshold required to trigger RASER. These diminishing bursts of *1–3d* or *3e* protons may be explained by the convection of the still hot solution[19] (the polarized sample outside of the coil mixes with the sample inside the coil, which additionally maintains the polarization over a longer time) as their polarization cannot be supplied directly by the chemical addition of *p*-H₂ at high field, unlike the polarization of the *b* protons. The longevity of *1–3d* or *3e* RASER signals is limited by their respective $T_1$ relaxation times (see Supplementary Table 4 for $T_1$ values) and initial polarization in the specific experiment. The RASER activity of the *c* protons is not always observed (e.g., in Fig. 4e, f); the *c* multiplet has both absorptive and emissive components (Supplementary Fig. 15), and whether this magnetization is enough for the RASER activity is dependent on molar polarization, which itself depends on the conversion of the substrate to the HP product.

In the case of **4** (Fig. 4k, l) and once with **1** (Supplementary Fig. 4), however, polarization is transferred in some other way and is strong enough to trigger continuous RASER signal of *4d* and *1d* protons, respectively. We assume that these protons are polarized via NOE—thus, positive polarization of the H$_A$ protons is converted into negative polarization of the recipient protons. The inconsistency of the *1d* RASER pattern across different experiments may be explained by the differences in chemical conversion— larger molar polarization of *1a* eventually leads to larger molar polarization of *1d*. Indeed, the only experiment where reinvigoration of *1d* RASER happens is the one where we have a significantly larger chemical conversion of **1'** to **1** (see Supplementary Table 5). In a similar fashion, initially negatively polarized *c* protons (as indicated by their presence in the ALTADENA RASER signals) experience NOE-mediated polarization transfer from the negatively polarized *b* protons. This leads to positive polarization and enhanced absorptive NMR signals of the c protons observed after the relaxation of the samples after ALTADENA RASER experiments (Supplementary Fig. 8).

## Simulations of RASER induced via NOE

To the best of our knowledge, previously, there were no reports of PHIP RASER induced via NOE observed here. Thus, we set ourselves to simulate this phenomenon to support our assumption. For this purpose, Solomon equations[44] were modified in a way to include the possibility of RASER (Eq. 3):

$$
\begin{cases}
\frac{d\langle \hat{I}_z \rangle}{dt} = -\rho_I \left( \langle \hat{I}_z \rangle - I_z^{eq} \right) - \sigma \left( \langle \hat{S}_z \rangle - S_z^{eq} \right) + \frac{\langle \hat{I}_+ \rangle^2}{|M_0 \tau_{RD}|} + \Gamma(t)\left( M_0 - \langle \hat{I}_z \rangle \right) \\
\frac{d\langle \hat{S}_z \rangle}{dt} = -\rho_S \left( \langle \hat{S}_z \rangle - S_z^{eq} \right) - \sigma \left( \langle \hat{I}_z \rangle - I_z^{eq} \right) + \frac{\langle \hat{S}_+ \rangle^2}{|M_0 \tau_{RD}|} \\
\frac{d\langle \hat{I}_+ \rangle}{dt} = -\nu_I \langle \hat{I}_+ \rangle - \frac{\langle \hat{I}_z \rangle \langle \hat{I}_+ \rangle}{|M_0 \tau_{RD}|} \\
\frac{d\langle \hat{S}_+ \rangle}{dt} = -\nu_S \langle \hat{S}_+ \rangle - \frac{\langle \hat{S}_z \rangle \langle \hat{S}_+ \rangle}{|M_0 \tau_{RD}|}
\end{cases}
$$

$$(3)$$

where $\langle \hat{I}_z \rangle$ and $\langle \hat{S}_z \rangle$ are the longitudinal magnetizations of the spins $I$ and $S$, respectively, $\langle \hat{I}_+ \rangle$ and $\langle \hat{S}_+ \rangle$ are their transverse magnetizations, $I_z^{eq}$ and $S_z^{eq}$ are the longitudinal magnetizations at thermal equilibrium, $\rho_i$ are the spin-lattice relaxation rates $(T_1^{-1})$, $\nu_i$ are the spin-spin relaxation rates $(T_2^{-1})$ and $\sigma$ is the cross-relaxation rate. $M_0$ is the initial magnetization of the system, $\tau_{RD}$ is the radiation damping time, $\Gamma(t)$ is the pumping rate expressed as $\alpha \cdot \exp(-t/\tau_p)$, where $\alpha$ is the pumping rate of $\langle \hat{I}_z \rangle$ and $\tau_p$ is the time of pumping decay.

The simulations indeed demonstrate the accumulation of negative longitudinal magnetization of the acceptor spin $S$ with its further transformation into transverse magnetization leading to RASER activity of this proton (Fig. 5). The evident difference in the time of $S$ signal appearance (ca.

5 s in the simulation vs. ca. 20 s in the real experiment) may be explained by the DDF of the real sample hindering dipole-dipole interactions, postponing the NOE-mediated polarization transfer. Further investigation proved that the magnetization pumping of spin $I$ is not required to produce multiple bursts of RASER of the spin $S$ as high polarization of $I$ is itself a pumping source (Supplementary Figs. 18 and 19), although this is slightly detrimental to the longevity of the RASER signal. Additionally, we demonstrate that continuous pumping of RASER-active spin $I$ leads to the preservation of positive polarization of the spin $S$, denying its possibility to appear in RASER, as is the case for the $c$ protons in ALTADENA RASER experiments (Supplementary Fig. 20). For further details, see Supplementary Note 11.

## Conclusions

In this work, we demonstrated multimodal PHIP RASER (containing more than 2 modes) using the example of hyperpolarized substituted allylic compounds. Overall, HP substances **1–4** obtained at PASADENA conditions produced various types of RASER, from monomodal to trimodal regimes, sometimes producing frequency combs. The emergence of RASER signals is interdependent as feedback of the RF-coil induces polarization transfer via NOE leading to continuous RASER activity of all vinyl protons. The importance of continuous interaction of the sample with the RF-coil during the bubbling of $p$-$H_2$ and concomitant polarization buildup for the intensity of PASADENA RASER and the number of produced modes was demonstrated. The RASER produced by the pairwise addition of $p$-$H_2$ inside the NMR probe tends to last for over 100 s.

ALTADENA RASER of the studied molecules revealed that the polarization spontaneously transferred to other $J$-coupled protons in the spin systems was enough to trigger the RASER of these nuclei. Generally, the RASER-active protons produced signals independently with the notable exception of allyl alcohol that on one occasion demonstrated intermittent RASER leading to continuous RASER with two active frequencies.

While the phenomena observed in both PASADENA and ALTADENA conditions were generally reproducible, the finer details of the RASER signals (order of appearance of the frequencies, appearance or absence of certain weak RASER signals, etc.) were found to be specific for each experiment. This is mainly influenced by varying chemical conversion/molar polarization; in PASADENA inside the RF-coil RASER experiments the chaotic evolution of magnetization also leads to a high degree of variability in RASER patterns.

Thus, we demonstrate the feasibility of inducing RASER on protons that do not originate from $p$-$H_2$ mediated via scalar couplings and intramolecular PRINOE interactions. These ways to induce RASER may prove to be useful in two-dimensional NMR spectroscopy techniques that allow to determine molecular structures employing scalar coupling networks and NOE. Since conventional pulse sequences may be detrimental to RASER activity, the implementation of pulse sequences similar to spin-noise-detected 2D NMR spectroscopy[45] might be considered. The concept of RASER induced by intramolecular polarization transfer via PRINOE in principle may be extrapolated to the intermolecular case—in this way, RASER of the analytes added to the sample may be induced, providing a new qualitative analytic instrument. Overall, the potent RASER of the investigated allylic compounds may prove useful in the quest to improve PRINOE-mediated intermolecular polarization transfer to other added analytes. We hope to report the results of these findings in the near future.

## Methods
### Sample preparation
Commercially available unsaturated substrates (propargyl alcohol (Sigma-Aldrich, 99%), 3-butyn-2-ol (Alfa Aesar, 98%), 2-methyl-3-butyn-2-ol (Sigma-Aldrich, 98%)), bis(norbornadiene)rhodium(I) tetrafluoroborate ([Rh(nbd)$_2$]BF$_4$, nbd = norbornadiene, Umicore, 99.9%), 1,4-bis(diphenylphosphino)butane (dppb, Sigma-Aldrich, 98%), methanol-$d_4$ (Zeotope, 99.8% D) and ultrapure hydrogen ( > 99.999%) were used as received. Propargyl pyruvate was synthesized according to previously published procedure[46]. At first, the mixture of [Rh(nbd)$_2$]BF$_4$ and dppb was

dissolved in methanol-$d_4$. The amounts of reactants were calculated to get the 10 mM concentration of Rh complex; dppb was taken in ~2% molar excess with respect to [Rh(nbd)$_2$]BF$_4$. The solution was left for ~30 min with periodic mixing to ensure formation of [Rh(nbd)(dppb)]BF$_4$ complex. The resultant solution (0.5 mL aliquots) was placed in standard 5 mm Wilmad NMR tubes tightly connected with ¼ in. outer diameter PTFE tubes. Before conducting the experiment, a portion of one of the unsaturated precursors **1'–4'** was added to an NMR tube; the amount of an unsaturated substrate was calculated so that its concentration in the resultant solution was 0.8 M (neglecting the slight dilution due to own volume of the added liquid). The solution was thoroughly mixed afterwards to ensure homogeneous distribution of the added substrate.

### Experimental considerations
Hydrogen gas was enriched with $p$-$H_2$ to ~94% content using parahydrogen generator based on a closed-cycle helium cryostat (CryoPribor, CFA-200-H2CELL) and cryo-compressor (Sumitomo, Zephyr HC-4A). Hydrated iron oxide FeO(OH) (Sigma-Aldrich, 371254) was used in a cell as a spin conversion catalyst. The scheme of the experimental setup is presented in Fig. 1b. The gas lines were purged with $p$-$H_2$ for ~1 min. Then the sample typically containing 10 mM [Rh(nbd)(dppb)]BF$_4$ catalyst, 10 mM free nbd (leftover from the reaction of [Rh(nbd)$_2$]BF$_4$ precursor with dppb), and 0.8 M of one of the precursors **1'–4'** dissolved in CD$_3$OD was connected to the gas lines. The sample was pressurized to 6.2 bar (regulated by a 75 psig safety valve) while the by-pass valve (see Fig. 1b) was opened. The gas flow rate was regulated using a mass flow controller (Brooks Instrument, model 5850E).

### ALTADENA RASER protocol
The sample was heated to 80 °C in a beaker with hot water for 30 s. Then $p$-$H_2$ bubbling was initiated by closing the by-pass valve; the gas was bubbled at 100 standard cubic centimeters per minute (sccm) gas flow rate for 15 s. Next, the sample was rapidly taken out of the beaker, dried with a paper towel, and placed into the NMR probe of a 7.05 T Bruker AV 300 NMR spectrometer. The sample transfer time was ~7 s. NMR signal acquisition was started several seconds before the sample was placed into the NMR probe.

### PASADENA inside the RF-coil RASER protocol
The sample was heated to 80 °C in a beaker with hot water for 30 s. Then the sample was rapidly taken out of the beaker, dried with a paper towel, and placed into the NMR probe of a 7.05 T Bruker AV 300 NMR spectrometer. The sample transfer time was ~7 s. Next, $p$-$H_2$ bubbling was initiated by closing the by-pass valve; the gas was bubbled at 100 sccm gas flow rate for 15 s. NMR signal acquisition was started immediately after the cessation of the $p$-$H_2$ flow (via opening the by-pass valve).

### PASADENA above the RF-coil RASER protocol
The sample was heated to 80 °C in a beaker with hot water for 30 s. Then the sample was rapidly taken out of the beaker, dried with a paper towel, and placed 7 cm above the NMR probe of a 7.05 T Bruker AV 300 NMR spectrometer. The measurements of the magnetic field profile along the bore of the NMR spectrometer confirmed that in this zone magnetic field is still the same as inside the NMR probe. The sample transfer time was ~7 s. Next, $p$-$H_2$ bubbling was initiated by closing the by-pass valve; the gas was bubbled at 100 sccm gas flow rate for 15 s. NMR signal acquisition was started simultaneously with the stop of the $p$-$H_2$ flow (via opening the by-pass valve) before the introduction of the sample into the NMR probe so that the whole RASER signal could be acquired.

### Acquisition of RASER signal
RASER signal was acquired without the application of an RF excitation pulse. A 5 mm PH DUL 300S1 C-H-D-05 CIDNP NMR probe with a $Q$ factor of ca. 550 was utilized. See Supplementary Methods for the exact parameters of the acquired spectra.

## Measurement of radiation damping time

The sample of 90% $H_2O$/10% $D_2O$ mixture (in the standard 5 mm NMR tube) with a 1/16" catheter inside (inserted to emulate the structure of the sample used in the RASER experiments) was prepared. An NMR spectrum of the sample was acquired by application of 90° RF pulse. The radiation damping time was calculated from FWHM of the $H_2O$ NMR signal according to the literature[47]:

$$\tau_{RD} = 0.8384/(\pi \cdot FWHM) = 44.3 \ ms$$

The coil quality factor $Q$ was estimated from the FWHM of the wobble curve as

$$Q = \omega_0/\Delta\omega = 300 \ MHz/0.54 \ MHz \approx 550$$

where $\omega_0$ is the frequency to which the RF-coil circuit was tuned and matched, $\Delta\omega$ is the FWHM of the wobble curve dip. From this, the filling factor η was estimated:

$$1/\tau_{RD} = \kappa\eta Q; \eta Q = 41 \rightarrow \eta \approx 0.075 = 7.5\%$$

For detailed information, see Supplementary Methods and Supplementary Note 3.

## Data availability

The data analyzed in this study are available at Zenodo data repository[48].

## Code availability

The NOE RASER simulation script is available at Zenodo data repository[48].

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

## Acknowledgements

O.G.S., A.P.Y., and I.V.K. thank the Russian Science Foundation (grant #22-43-04426) for the support of all PHIP RASER experiments. I.A.T., H.d.M., and A.B.S. acknowledge support by the BMBF in the funding program "Quantum Technologies—from Basic Research to Market" under the project "QuE-MRT" (contract number: 13N16448), the German Cancer Consortium (DKTK), the DKTK Joint Funding project "HYPERBOLIC", the Research Commission of the University Medical Center Freiburg, B.E.S.T. Fluidsysteme GmbH I Swagelok Stuttgart, and the DFG (#SCHM 3694/1, #SCHM 3694/2, #SFB1479 Project ID: 441891347SFB1160). A.N.P. and J.B.H. acknowledge funding from German Federal Ministry of Education and Research (BMBF) within the framework of the e: Med research and funding concept (01ZX1915C), DFG (PR 1868/3-1, PR 1868/5-1, HO-4602/2-2, HO-4602/3, EXC2167, FOR5042, TRR287). MOIN CC was founded by a grant from the European Regional Development Fund (ERDF) and the Zukunftsprogramm Wirtschaft of Schleswig-Holstein (Project no. 122-09-053). E.Y.C. thanks the following for funding: National Science Foundation grant: CHE-1904780, NIBIB R01EB034197, NHLBI 1R21HL154032, and DOD CDMRP W81XWH-20-10576. E.Y.C. and A.B.S. thank Wayne State University for a Postdoctoral Fellow award.

## Author contributions

PHIP RASER experiments: I.A.T., O.G.S., and A.P.Y. Data processing: I.A.T., O.G.S., and H.d.M. RASER simulations: I.A.T. and H.d.M. Supervision: O.G.S. and A.B.S. Funding: I.V.K. and O.G.S. Original draft preparation: I.A.T. and O.G.S. Data interpretation, discussion, paper proof-reading: all authors (including A.N.P., E.Y.C., and J.B.H.). This work was initiated in May 2022. I.A.T. performed experiments at International Tomography Center SB RAS (Novosibirsk, Russia) and wrote the manuscript, processed the data and performed simulations at University Medical Center Freiburg (Germany).

## Funding

## Competing interests

E.Y.C. declares a stake of ownership in XeUS Technologies, Ltd. E.Y.C. also holds stock of Vizma Life Sciences (VLS) and serves on the scientific advisory board (SAB) of VLS. The remaining authors declare no competing interests.
