## [Peer Review File · Communications Chemistry]

Through-Bond and Through-Space Radiofrequency Amplification by Stimulated Emission of Radiation

Corresponding Author: Mr Ivan Trofimov

Version 0:

Reviewer comments:

Reviewer #1

(Remarks to the Author)

In this work Trofimov et al. report various types of RASER effect on different substrates. The experiments are carried out according to the ALTADENA and PASADENA protocols. At high magnetic fields, two different modalities are investigated: RASER inside the probe and RASER above the probe.

The types of NMR patterns observed are substrate-specific and not always reproducible, as the authors acknowledge, showing the delicate interplay between spin dynamics and experimental conditions. Nonetheless, the paper is quite thorough in examining all the different contributions in a systematic manner.

The paper is publishable.

There are some points, however, that should be further clarified. The presence of c) signals in PASADENA RASER (for instance, in 4)) is attributed to intramolecular NOE relaxation. It seems reasonable. It is not so clear why c) is not always detected (in ALTADENA RASER 2) if I interpret the figures correctly). Why?

On the experimental protocol: Is the parahydrogen pressure released after the bubbling? If not, would you expect to still detect long-lived RASER effects if the H₂ gas is released after bubbling? In the ALTADENA RASER, why the b signals seem to be long-lived and the d signals are not?

The intermittent nature of some of the RASER signals is a bit obscure and I wouldn't know how this can be useful for 2D experiments like COSY and NOESY. However, I agree that the findings are interesting and would need further elucidation in future investigations.

The paper is well-written, quite accurate, and well documented. I suggest publication after the points above are addressed.

Reviewer #2

(Remarks to the Author)

The document explores the phenomenon of Radio Amplification by Stimulated Emission of Radiation (RASER) in the context of nuclear magnetic resonance (NMR) experiments with hyperpolarized substances. Specifically, the study focuses on Para-Hydrogen-Induced Polarization (PHIP) RASER induced via Nuclear Overhauser Effect (NOE). The experiments involve pairwise addition of parahydrogen to propargylic compounds, resulting in RASER activity beyond the chemically introduced protons. By simulating RASER induced via NOE interactions, the study demonstrates the accumulation of negative longitudinal magnetization leading to subsequent RASER activity, supporting the experimental results. Additionally, the study investigates the longevity and dynamics of RASER signals, shedding light on the mechanisms. Through detailed experimental protocols and simulations, the study provides insights into the interplay of chemical conversion, polarization transfer, and NOE effects in inducing and sustaining RASER in hyperpolarized systems.). The "PR-inside" experiments show the long-lived RASER of vinyl protons, and illustrate the impact of experimental conditions on the appearance of RASER emissions. The observation of RASER emission consecutive to NOE transfer is a major discovery of this work.

In the introduction (page 2, §2) there is very useful comprehensive summary of PHIP principles, which will surely be appreciated by a broader readership.

Here are some detailed comments and questions, which might help to improve the manuscript slightly:

The definition of T_2^* (after Eq. 2) uses " $\gamma\Delta B_0$ " to describe the inhomogeneous broadening contribution. I am not sure if this is also true for non-linear field gradients. On page 2 §1 it might be worth noting that (non-continuous) RASER has been achieved by inversion of high thermal polarization (e.g. Chemical Physics Letters 623, 55-59, 2015).

When referring to the different types of PHIP-MASER experiments I notice some inconsistency between the terms used in Figure 1 and the text (on page 3) referring to it. In view of the importance of coupling to the rf-circuit, I'd suggest the authors use "above/inside rf-coil" instead of "above/inside probe" and also PR-above/PR-inside..

Concerning the deviation of the frequencies from expected ones and drift over time, the authors might also consider effects of frequency pushing (pulling) observed in presence of strong polarization. They do mention DDF effects on page 5 but the conceptually simpler frequency pushing/pulling might also be considered here. (Gueron, Magn. Res. In Medicine 19, 31-41 (1991))

It might be interesting to use a Q-switched probe (which was somewhat popular in the early 2000s) to change the RASER condition reproducibly during these experiments. This might also help to distinguish between NOE and Hartmann-Hahn type transfer. (page 4)

The PHIP RASER NOE simulations support the idea of an intramolecular NOE impressively, but they do not completely rule out other mechanisms. The authors indicate that these phenomena are still under investigation.

Minor remarks:

Sometimes the abbreviation SI is used and sometimes ESI for the Supplement.

In view of the importance of the radiation damping rate for the RASER phenomenon (Eq. 1) (an estimate of) the filling factor would be interesting experimental detail and should be given in the Methods section, together with Q.

There are few language problems, in particular in the supplementary information, (some missing articles, usage of adverbs), which should be taken care of by a natively English speaking editor.

Summary:

It is an excellent timely article addressing important questions and introducing innovative methods with potentially high impact on future research and applications of para-hydrogen and hyperpolarization in NMR. The relevant literature has been cited adequately.

I recommend acceptance of the manuscript after minor amendments as suggested above.

Reviewer #3

(Remarks to the Author)

In this work, the authors investigate the RASER effect induced by PHIP of various terminal alkynes to give hyperpolarised terminal alkenes. They highlight the novelty of observing the RASER effect for 3 or more spins, in contrast to previous observations of bimodal RASER signals from PHIP. The work is thoroughly described and should be published. However, I have some comments on the claims of novelty and applications. The work will be of interest to researchers exploring the PHIP RASER effect, who are predominantly within the chemistry community.

1. I interpret the prefix "multi" to mean "multiple", i.e., "more than one". So I disagree with the claim of demonstrating "the first multimodal PHIP RASER". The previous bimodal RASER experiments are also multimodal.

2. The authors state "To the best of our knowledge, previously, there were no reports of PHIP RASER induced via NOE observed here." Perhaps I am misunderstanding this, but there are multiple descriptions of this in the literature (PRINOE), which the authors cite in their introduction? E.g. 10.1002/anie.202108306.

3. The authors mention in the abstract and conclusion that the RASER effect can be used in correlation experiments such as COSY and NOESY. I disagree with this claim, since these are multi-pulse sequences that are incompatible with the spontaneous RASER effect. Without proof of the feasibility of these experiments, I believe such claims should be removed.

Minor comments:

4. The authors mention that "RASER applied on two protons can fulfill Hartmann-Hahn conditions". Can the authors provide a full explanation of this, or relevant literature references?

5. The authors mention that "during PR-inside experiments, the nascent polarization of the sample interacts with the coil during polarization buildup and before the start of acquisition". However, before acquisition the receiver is blanked and therefore there is no radiation damping. What interaction do the authors expect with the coil in this case?

6. In the data availability statement, the authors state the data is included in the article and SI, presumably referring to the data presented in the figures. In my view, the data is not available in this format. The authors should consider uploading the data to a repository, or correct the statement that the data is not available.

Version 1:

Reviewer comments:

Reviewer #2

(Remarks to the Author)

The answers provided by the authors and the revised manuscript adequately address all issues brought up by the reviewers. I also would like to commend the authors for the very detailed supplementary information provided.

I might suggest a minor improvement to Figure 5, which also applies to the Supplementary Figures 18, 19 and 20: The grey boxes indicating the NMR-coil in (a) could be extended to include the RASER part of the schemes, as the NMR-coil is essential for RASER emissions to occur.

Assuming that formatting/typesetting issues as well as minor inconsistencies in the references style will be taking care of in the publishing process the manuscript is recommended for publication as is.

Reviewer #3

(Remarks to the Author)

The authors have addressed my comments and the manuscript should be accepted. I have also reviewed the responses to Reviewer 1 and believe that these points have also been addressed.

A couple of small comments to the authors:

"polarization transfer from b to c protons akin to microwave irradiation triggering homo- and heteronuclear polarization transfer" - do the authors mean radiofrequency irradiation, not microwave?

Regarding point 5 (the Q factor of the circuit when the receiver is not on). The observation that the authors "missed" the start of the RASER by starting acquisition too late is convincing evidence that the circuit is tuned before acquisition. This may be spectrometer dependent. The authors might be interested to know that using a Bruker AVANCE III spectrometer, the DNP-induced RASER effect was reliably initiated by turning on acquisition, indicating that the radiation damping was insufficient before acquisition (10.1021/acs.jpcllett.0c03457). This was using a conventional probe, not Q switched, the "switching" came from the spectrometer. Another difference may be that the negative hyperpolarisation was generated during the recycle delay of the pulse sequence, i.e. while the sequence was running - this could also affect whether the receiver is blanked or not.

Pointwise Response to the Referees

Through-Bond and Through-Space Radiofrequency Amplification by Stimulated Emission of Radiation

Reviewer #1

In this work Trofimov et al. report various types of RASER effect on different substrates. The experiments are carried out according to the ALTADENA and PASADENA protocols. At high magnetic fields, two different modalities are investigated: RASER inside the probe and RASER above the probe.

The types of NMR patterns observed are substrate-specific and not always reproducible, as the authors acknowledge, showing the delicate interplay between spin dynamics and experimental conditions. Nonetheless, the paper is quite thorough in examining all the different contributions in a systematic manner.

The paper is publishable.

Author Reply: We thank the Reviewer for their enthusiasm about our manuscript. To additionally address the topic of reproducibility, we added a paragraph about it to the Conclusions.

Changes made to the manuscript: The following paragraph was added to the Conclusions section: “While the phenomena observed in both PASADENA and ALTADENA conditions were generally reproducible, the finer details of the RASER signals (order of appearance of the frequencies, appearance or absence of certain weak RASER signals, etc.) were found to be specific for each experiment. This is mainly influenced by varying chemical conversion/molar polarization; in PASADENA inside the RF-coil RASER experiments the chaotic evolution of magnetization also leads to a high degree of variability in RASER patterns.”

There are some points, however, that should be further clarified. The presence of c) signals in PASADENA RASER (for instance, in 4)) is attributed to intramolecular NOE relaxation. It seems reasonable. It is not so clear why c) is not always detected (in ALTADENA RASER 2) if I interpret the figures correctly). Why?

Author Reply: We thank the Reviewer for this question. The initial polarization of the c protons in ALTADENA experiments is determined by the J-coupling network parameters. As suggested by the simulations provided in the SI, some components of the c signals are emissive—in case of sufficiently strong intensity, they likely contribute to the initial weak c signals visible in the presented ALTADENA RASER spectrographs. If this initial polarization of the c protons is not large enough (e.g., due to low conversion or inefficient polarization transfer), the RASER is not initiated. After the placement of the sample inside the coil, NOE-mediated transfer of polarization from the H_B (b) to the c protons starts happening. As we work with small molecules, this leads to the conversion of negative polarization of the b to positive polarization of the c protons (as may be seen in the spectra of the relaxing samples after ALTADENA RASER experiments in Supplementary Figure 8), which makes their contribution to the RASER signal impossible. We additionally performed a simulation of this case, the results of which are now provided in the SI.

Changes made to the manuscript: The following text was added on page 5, in the end of §8: “The RASER activity of the c protons is not always observed; the c multiplet has both absorptive and emissive components (Supplementary Figure 15), and whether this magnetization is enough for the RASER activity is dependent on molar polarization, which itself depends on the conversion of the substrate to the HP product.”

The following text was added on page 6, in the end of §1: “In a similar fashion, initially negatively polarized c protons (as indicated by their presence in the ALTADENA RASER signals) experience NOE-mediated polarization transfer from the negatively polarized b protons. This leads to Additionally, we demonstrate that continuous pumping of RASER-active spin I leads to the preservation of positive polarization of the spin S, denying its possibility to appear in RASER, as is the case for the c protons in ALTADENA RASER experiments (Supplementary Figure 20). positive

polarization and enhanced absorptive NMR signals of the c protons observed after the relaxation of the samples after ALTADENA RASER experiments (Supplementary Figure 8).”

A correction was made to the Eq. 3: M_{OTRD} term was corrected to $|M_{OTRD}|$.

Also, on page 7, §2 we added the following sentence: “Additionally, we demonstrate that continuous pumping of RASER-active spin I leads to positive polarization of the spin S, denying its possibility to appear in RASER, as is the case for the c protons in ALTADENA RASER experiments (Supplementary Figure 20).”

Supplementary Section 12 was expanded to include a new Supplementary Figure 20 and the accompanying text covering the results of the aforementioned simulation.

On the experimental protocol: Is the parahydrogen pressure released after the bubbling? If not, would you expect to still detect long-lived RASER effects if the H₂ gas is released after bubbling?

Author Reply: We thank the Reviewer for these questions. There is no pressure release after the stop of bubbling and during the acquisition of RASER. If we release the p-H₂ gas we would expect to observe shorter RASER signals, as ongoing hydrogenation reaction and, consequently, the replenishment of negative polarization will slow down due to p-H₂ pressure dropping from 6.2 to only 1 bar.

Changes made to the manuscript: The following sentence was added on page 4, in the end of §3: “Thus, the ongoing process of hydrogenation is quite important for the longevity of the obtained RASER signals.”

In the ALTADENA RASER, why the b signals seem to be long-lived and the d signals are not?

Author Reply: We thank the Reviewer for the question. For the b protons, negative polarization is directly replenished by addition of p-H₂ to the unreacted precursor, as discussed in the text. The case of the d protons is different: their initial polarization, induced in ALTADENA conditions via J-coupling networks, happens to be negative, thus, they contribute to the RASER signal. During the acquisition, the sample is in high field and thus replenishment of negative polarization of the d protons inside the coil is possible only via convection or NOE; the latter will maintain RASER of the d protons only if the positive molar polarization of the H_A (a) protons is large enough, as suggested by comparing the conversion of substrate (as observed in the RASER spectrograph in Figure S4 compared to Figure 4c). In case of convection, the limit would be determined by T₁ of the specific nuclei.

Changes made to the manuscript: We added the following text on page 6, in §5: “<...>, unlike the polarization of the b protons. The longevity of 1–3d or 3e RASER signals is limited by their respective T₁ relaxation times (see Supplementary Table 4 for T₁ values) and initial polarization in the specific experiment.”

The reference to Supplementary Table 4 in the caption of Figure 5 was hence removed.

The intermittent nature of some of the RASER signals is a bit obscure and I wouldn't know how this can be useful for 2D experiments like COSY and NOESY. However, I agree that the findings are interesting and would need further elucidation in future investigations.

Author Reply: We thank the Reviewer for this comment. RASER signal, if maintained for a long duration of time, yields very narrow NMR signals after performing Fourier transformation. The idea of combining RASER with correlation NMR spectroscopy techniques actually is similar to spin-noise 2D NMR spectroscopy idea (10.1021/jz402100g).

Changes made to the manuscript: In the Conclusions, instead of:

“These ways to induce RASER may prove to be useful in NMR techniques such as COSY and NOESY that allow to determine molecular structures employing scalar coupling networks and NOE, respectively.”

the following text was added:

“These ways to induce RASER may prove to be useful in two-dimensional NMR spectroscopy techniques that allow to determine molecular structures employing scalar coupling networks and

NOE. Since conventional pulse sequences may be detrimental to RASER activity, implementation of pulse sequences similar to spin-noise-detected 2D NMR spectroscopy⁴⁴ might be considered.”

The paper is well-written, quite accurate, and well documented. I suggest publication after the points above are addressed.

Author Reply: We thank the Reviewer for their enthusiasm about our manuscript.

Reviewer #2

The document explores the phenomenon of Radio Amplification by Stimulated Emission of Radiation (RASER) in the context of nuclear magnetic resonance (NMR) experiments with hyperpolarized substances. Specifically, the study focuses on Para-Hydrogen-Induced Polarization (PHIP) RASER induced via Nuclear Overhauser Effect (NOE). The experiments involve pairwise addition of parahydrogen to propargylic compounds, resulting in RASER activity beyond the chemically introduced protons. By simulating RASER induced via NOE interactions, the study demonstrates the accumulation of negative longitudinal magnetization leading to subsequent RASER activity, supporting the experimental results. Additionally, the study investigates the longevity and dynamics of RASER signals, shedding light on the mechanisms. Through detailed experimental protocols and simulations, the study provides insights into the interplay of chemical conversion, polarization transfer, and NOE effects in inducing and sustaining RASER in hyperpolarized systems.). The “PR-inside” experiments show the long-lived RASER of vinyl protons, and illustrate the impact of experimental conditions on the appearance of RASER emissions. The observation of RASER emission consecutive to NOE transfer is a major discovery of this work.

Author Reply: We thank the Reviewer for their enthusiasm about our manuscript.

In the introduction (page 2, §2) there is very useful comprehensive summary of PHIP principles, which will surely be appreciated by a broader readership.

Here are some detailed comments and questions, which might help to improve the manuscript slightly:

The definition of $T2^*$ (after Eq. 2) uses “ $\gamma\Delta B_0$ ” to describe the inhomogeneous broadening contribution. I am not sure if this is also true for non-linear field gradients.

Author Reply: We thank the Reviewer for this comment. The definition of $T2^*$ is $1/T2 + 1/T2i = 1/T2 + \gamma\Delta Bi$, where ΔBi is a contribution of field inhomogeneities across the voxel. For an arbitrary shape of gradient $T2^*$ would vary across the sample. Hence, it would be more proper to note in the text that “ $\gamma\Delta B_0$ ” is correct for the case of linear field gradients.

Changes made to the manuscript: The following text was added on page 1, §3: “(in the case of linear field gradients)”.

On page 2 §1 it might be worth noting that (non-continuous) RASER has been achieved by inversion of high thermal polarization (e.g. Chemical Physics Letters 623, 55-59, 2015).

Author Reply: We agree with the Reviewer that it is worth to mention this method of producing RASER.

Changes made to the manuscript: The following text was added on page 2, in the end of §1: “RASER emissions may be also obtained by inversion of thermal polarization provided it is sufficiently high.^{27,28}”

When referring to the different types of PHIP-MASER experiments I notice some inconsistency between the terms used in Figure 1 and the text (on page 3) referring to it. In view of the importance

of coupling to the rf-circuit, I'd suggest the authors use "above/inside rf-coil" instead of "above/inside probe" and also PR-above/PR-inside..

Author Reply: We thank the Reviewer for this suggestion. It may indeed be better to refer to these methods as "PASADENA inside/above the RF-coil RASER". The abbreviations "PR-inside/above" are introduced in order to reduce the text length and make them more distinguishable in the text, where "PR" stands for "PASADENA RASER".

Changes made to the manuscript: In the caption of Figure 1 and in Figures 1–3 and S3 mentioning of "the probe" is changed to "the RF-coil". In Figure 1 the abbreviations "PR-inside/above" are now added near "PASADENA inside/above the RF-coil RASER".

Concerning the deviation of the frequencies from expected ones and drift over time, the authors might also consider effects of frequency pushing (pulling) observed in presence of strong polarization. They do mention DDF effects on page 5 but the conceptually simpler frequency pushing/pulling might also be considered here. (Gueron, Magn. Res. In Medicine 19, 31-41 (1991))

Author Reply: We thank the Reviewer for this suggestion and the provided reference. While this effect indeed has an influence on the positions of signals, the direction of the push/pull of the NMR signal is defined by its position relative to the frequency to which the RF-coil was tuned. The coil was manually tuned using WOBB before the RASER experiments, the resolution in the observed WOBB image is 10 kHz (~33 ppm at 7.05 T).

According to the formula provided in the referenced article, $(\omega_p - \omega_s)/\delta_s = (\omega_s - \omega_c)/\Delta_c$; typical drift of the NMR signals in the RASER experiments $(\omega_p - \omega_s) \sim 50$ Hz (e.g., in Supplementary Figure 3), $\delta_s \sim 5$ Hz, and $\Delta_c = 2\text{FWHM} \sim 1$ MHz.

Assuming the whole drift is explained by the pushing/pulling, we obtain $(\omega_s - \omega_c) \sim (\omega_p - \omega_s)/\delta_s * \Delta_c = (50 \text{ Hz}/5 \text{ Hz}) * 1 \text{ MHz} = 10 \text{ MHz}$, which would mean that we are far off the central RF-coil frequency and this is not the case.

Realistically, we may assume that over some time we have a central frequency drift of ± 100 kHz; then, $(\omega_p - \omega_s) \sim (\omega_s - \omega_c)/\Delta_c * \delta_s = (100 \text{ kHz}/1 \text{ MHz}) * 5 \text{ Hz} = 0.5 \text{ Hz}$. This amounts to ~1 % of the observed frequency drift.

Hence, we consider the frequency pushing/pulling effect, in this case, weak compared to the DDF effects and it is also unclear in which direction it would move the positions of the frequencies (due to ± 33 ppm precision in the tuning/matching program the central frequency may end up being either larger or smaller than the Larmor frequencies of the ^1H nuclei), while we consistently observe an initial downfield frequency shift due to the DDF—thus, we do not think it is necessary to discuss this effect in the manuscript.

Changes made to the manuscript: None.

It might be interesting to use a Q-switched probe (which was somewhat popular in the early 2000s) to change the RASER condition reproducibly during these experiments. This might also help to distinguish between NOE and Hartmann-Hahn type transfer. (page 4)

Author Reply: We thank the Reviewer for this suggestion.

Changes made to the manuscript: To elaborate on the point of importance of presence of p-H₂, they are now addressed in page 4 §2: "To establish the exact nature of the occurring polarization transfer experiments involving a Q-switchable NMR probe may be conducted."

The PHIP RASER NOE simulations support the idea of an intramolecular NOE impressively, but they do not completely rule out other mechanisms. The authors indicate that these phenomena are still under investigation.

Author Reply: We thank the Reviewer for this comment. We would like to point out that we expanded the main text and Supplementary Section 12 of the SI by an additional simulation that takes into consideration a pair of terminal vinyl spins b and c in ALTADENA RASER conditions. The resulting positive polarization of spin c in the end of the simulation period further supports the intramolecular NOE mechanism.

Changes made to the manuscript: On page 7, §2 we added the following sentence: “Additionally, we demonstrate that continuous pumping of RASER-active spin I leads to positive polarization of the spin S, denying its possibility to appear in RASER, as is the case for the c protons in ALTADENA RASER experiments (Supplementary Figure 20).”.

A correction was made to the Eq. 3: M_{0TRD} term was corrected to $|M_{0TRD}|$.

Supplementary Section 12 in the SI was expanded to include a new Supplementary Figure 20 and the accompanying text covering the results of the aforementioned simulation.

Minor remarks:

Sometimes the abbreviation SI is used and sometimes ESI for the Supplement.

Author Reply: We thank the Reviewer for this comment.

Changes made to the manuscript: The style of the referencing the Supplement Sections, Figures and Tables was corrected according to the Style formatting guidelines.

In view of the importance of the radiation damping rate for the RASER phenomenon (Eq. 1) (an estimate of) the filling factor would be interesting experimental detail and should be given in the Methods section, together with Q.

Author Reply: We thank the Reviewer for this suggestion. We agree that these parameters are quite important, however, in view of the length of the manuscript, we provide the method and the obtained values without expanded calculations in the Methods section. The full calculation is still available in the Supplementary Information file.

Changes made to the manuscript: In the Methods section, the suggested information was added.

There are few language problems, in particular in the supplementary information, (some missing articles, usage of adverbs), which should be taken care of by a natively English speaking editor.

Author Reply: We thank the Reviewer for this notice.

Changes made to the manuscript: The texts of the manuscript and the Supplementary Information were checked, found mistakes were corrected.

Summary:

It is an excellent timely article addressing important questions and introducing innovative methods with potentially high impact on future research and applications of para-hydrogen and hyperpolarization in NMR. The relevant literature has been cited adequately.

I recommend acceptance of the manuscript after minor amendments as suggested above.

Author Reply: We thank the Reviewer for their enthusiasm about our manuscript.

Reviewer #3

In this work, the authors investigate the RASER effect induced by PHIP of various terminal alkynes to give hyperpolarised terminal alkenes. They highlight the novelty of observing the RASER effect for 3 or more spins, in contrast to previous observations of bimodal RASER signals from PHIP. The work is thoroughly described and should be published.

Author Reply: We thank the Reviewer for their enthusiasm about our manuscript.

However, I have some comments on the claims of novelty and applications. The work will be of interest to researchers exploring the PHIP RASER effect, who are predominantly within the chemistry community.

1. I interpret the prefix “multi” to mean “multiple”, i.e., “more than one”. So I disagree with the claim of demonstrating “the first multimodal PHIP RASER”. The previous bimodal RASER experiments are also multimodal.

Author Reply: We thank the Reviewer for this comment. Indeed, a formal definition of a word “multiple” is “more than one”. However, in certain fields of physics, the distinction is made between the cases of one, two and more interacting objects. Here, we meant “multi-” as “more than two”. In order to clear up the ambiguity, we added an explanation near the word “multimodal.”

Changes made to the manuscript: The first sentence in the Conclusions section was changed to: “In this work, we demonstrated the first multimodal PHIP RASER (containing more than 2 modes) using the example of hyperpolarized substituted allylic compounds.”

2. The authors state “To the best of our knowledge, previously, there were no reports of PHIP RASER induced via NOE observed here.” Perhaps I am misunderstanding this, but there are multiple descriptions of this in the literature (PRINOE), which the authors cite in their introduction? E.g. 10.1002/anie.202108306.

Author Reply: We thank the Reviewer for this notice. In the work by Korchak et al. (10.1002/anie.202108306), indeed, RASER is utilized to induce NOE—however, the resultant NMR signal enhancement of the recipient molecules was not enough to trigger RASER. Here, we show that RASER of the recipient nuclei may be induced via NOE in PHIP experiments (NOE induces RASER vs. RASER induces NOE), which is a novel part of this work.

Changes made to the manuscript: None suggested.

3. The authors mention in the abstract and conclusion that the RASER effect can be used in correlation experiments such as COSY and NOESY. I disagree with this claim, since these are multi-pulse sequences that are incompatible with the spontaneous RASER effect. Without proof of the feasibility of these experiments, I believe such claims should be removed.

Author Reply: We thank the Reviewer for this comment. Indeed, approaching the RASER signal with conventional correlation spectroscopy techniques would most likely not result in a success. However, it was shown possible to extract 2D spectral information from spin noise (e.g., snHMQC in 10.1021/jz402100g). Similarly, one may specifically modify the existing pulse sequences to comply with the RASER limitations. In PHIP RASER conditions, as described in our manuscript, this may not be the best approach as our substrate depletes (even then, one of the first PHIP RASER papers, 10.1002/cphc.201901056, describes a ~10 min long RASER signal acquisition during p-H₂ bubbling). As for the homonuclear correlation spectroscopy techniques, it would certainly be more challenging to modify them—however, RF pulses are not necessarily detrimental to RASER activity, and may in fact trigger RASER emissions.

Thus, we believe that combining the RASER effect with two-dimensional spectroscopy techniques (not necessarily COSY or NOESY) is feasible. In order to make the way we believe it may be done clearer, we expanded and corrected the relevant section in the Conclusions.

Changes made to the manuscript: In the Abstract, instead of “...correlation NMR spectroscopy techniques” the following text was added: “...two-dimensional NMR spectroscopy techniques”.

In the Conclusions, instead of:

“These ways to induce RASER may prove to be useful in NMR techniques such as COSY and NOESY that allow to determine molecular structures employing scalar coupling networks and NOE, respectively.”

the following text was added:

“These ways to induce RASER may prove to be useful in two-dimensional NMR spectroscopy techniques that allow to determine molecular structures employing scalar coupling networks and NOE. Since conventional pulse sequences may be detrimental to RASER activity, implementation of pulse sequences similar to spin-noise-detected 2D NMR spectroscopy⁴⁵ might be considered.”

Minor comments:

4. The authors mention that “RASER applied on two protons can fulfill Hartmann-Hahn conditions”. Can the authors provide a full explanation of this, or relevant literature references?

Author Reply: We thank the Reviewer for this question. In that paragraph, we discuss two possibilities of how the polarization transfer from H_B to H_C may occur: it may be either via NOE or via Hartmann-Hahn condition. By the latter we mean that the RASER emissions produced by HP molecules in the solution may trigger polarization transfer from H_B to H_C—akin to microwave irradiation triggering ¹H–X polarization transfer in solid and liquid states.

Changes made to the manuscript: The following text was added on page 4, §2: “<...>³⁹–RASER emissions produced by HP molecules in the solution may trigger polarization transfer from b to c protons akin to microwave irradiation triggering homo- and heteronuclear polarization transfer.⁴⁰”.

5. The authors mention that “during PR-inside experiments, the nascent polarization of the sample interacts with the coil during polarization buildup and before the start of acquisition”. However, before acquisition the receiver is blanked and therefore there is no radiation damping. What interaction do the authors expect with the coil in this case?

Author Reply: We thank the Reviewer for this question. In our understanding, when no NMR signals are acquired, the receiver is definitely blanked, however, it does not affect the tuned circuit of the NMR probe and its Q-factor. E.g., the receiver is blanked during the application of an RF pulse in conventional NMR experiments as well—but the RF coil has to be fully operational during this time. Also, it would require a probe with Q-switching capabilities to change the RASER conditions while the sample resides inside the sensitive zone of the RF coil. Thus, there may still be radiation damping and concomitant RASER activity during the bubbling inside the RF-coil before the signal acquisition is initiated. In fact, in some of the preliminary experiments (not shown in the manuscript) the NMR signal acquisition was started too late, so we observed the RASER oscillations right in the beginning of the acquired signal.

Changes made to the manuscript: None suggested.

6. In the data availability statement, the authors state the data is included in the article and SI, presumably referring to the data presented in the figures. In my view, the data is not available in this format. The authors should consider uploading the data to a repository, or correct the statement that the data is not available.

Author Reply: We thank the Reviewer for this notice.

Changes made to the manuscript: Data availability statement is changed to: “The data analyzed in this study and the RASER simulation script are available at Zenodo data repository.⁴⁸”

Pointwise Response #2 to the Referees

Through-Bond and Through-Space Radiofrequency Amplification by Stimulated Emission of Radiation

Reviewer #2 (Remarks to the Author):

The answers provided by the authors and the revised manuscript adequately address all issues brought up by the reviewers.

I also would like to commend the authors for the very detailed supplementary information provided.

Author Reply: We thank the Reviewer very much for this comment.

I might suggest a minor improvement to Figure 5, which also applies to the Supplementary Figures 18, 19 and 20: The grey boxes indicating the NMR-coil in (a) could be extended to include the RASER part of the schemes, as the NMR-coil is essential for RASER emissions to occur.

Author Reply: We thank the Reviewer for this suggestion.

Changes made to the manuscript: In Figures 5 and Supplementary Figures 18–20 the grey boxes were extended. Also, in order to keep terms used in the manuscript consistent, all instances of “NMR coil” were renamed to “RF-coil”, in the text and in the Figures.

Assuming that formatting/typesetting issues as well as minor inconsistencies in the references style will be taken care of in the publishing process the manuscript is recommended for publication as is.

Author Reply: We thank the Reviewer for their enthusiasm about our manuscript.

Reviewer #3 (Remarks to the Author):

The authors have addressed my comments and the manuscript should be accepted. I have also reviewed the responses to Reviewer 1 and believe that these points have also been addressed.

Author Reply: We thank the Reviewer for their enthusiasm about our manuscript.

A couple of small comments to the authors:

"polarization transfer from b to c protons akin to microwave irradiation triggering homo- and heteronuclear polarization transfer" - do the authors mean radiofrequency irradiation, not microwave?

Author Reply: We thank the Reviewer for this notice. Indeed, RF-irradiation was meant here.

Changes made to the manuscript: The instance of “microwave irradiation” was changed to “RF-irradiation”.

Regarding point 5 (the Q factor of the circuit when the receiver is not on). The observation that the authors "missed" the start of the RASER by starting acquisition too late is convincing evidence that the circuit is tuned before acquisition. This may be spectrometer dependent. The authors might be interested to know that using a Bruker AVANCE III spectrometer, the DNP-induced RASER effect was reliably initiated by turning on acquisition, indicating that the radiation damping was insufficient before acquisition (10.1021/acs.jpcclett.0c03457). This was using a conventional probe, not Q switched, the "switching" came from the spectrometer. Another difference may be that the negative hyperpolarisation was generated during the recycle delay of the pulse sequence, i.e. while the sequence was running - this could also affect whether the receiver is blanked or not.

Author Reply: We thank the Reviewer for this interesting notice.

Author Reply: In the end, we would like to thank all Reviewers for their questions, suggestions and critique. We believe that the process of response to the points that were raised led to improvement of quality and accessibility of our manuscript.